# ADDER: ADAPTED DENSE RETRIEVAL

## ABSTRACT

Information retrieval involves selecting artifacts from a corpus that are most relevant to a given search query. The flavor of retrieval typically used in classical applications can be termed as *homogeneous* and *relaxed*, where queries and corpus elements are both natural language (NL) utterances (homogeneous) and the goal is to pick most relevant elements from the corpus in the Top-K, where K is large, such as 10, 25, 50 or even 100 (relaxed). Recently, retrieval is being used extensively in preparing prompts for large language models (LLMs) to enable LLMs to perform targeted tasks. These new applications of retrieval are often *heterogeneous* and *strict* – the queries and the corpus contain different kinds of entities, such as NL and code, and there is a need for improving retrieval at Top-K for small values of K, such as K=1 or 3 or 5. Current dense retrieval techniques based on pretrained embeddings provide a general-purpose and powerful approach for retrieval, but they are oblivious to task-specific notions of similarity of heterogeneous artifacts. We introduce *Adapted Dense Retrieval*, a mechanism to transform embeddings to enable improved task-specific, heterogeneous and strict retrieval. Adapted Dense Retrieval works by learning a low-rank residual adaptation of the pretrained black-box embedding. We empirically validate our approach by showing improvements over the state-of-the-art general-purpose embeddings-based baseline.

## 1 INTRODUCTION

Information retrieval has a long and diverse history. A variety of approaches have been proposed (Turtle & Croft, 1989; Crestani et al., 1998; Cao et al., 2006; Akkalyoncu Yilmaz et al., 2019b; Ye et al., 2016), yet retrieval continues to remain a challenging problem. The goal of retrieval is simple: extract the most relevant artifacts from a large corpus given a query. Retrieval approaches are broadly classified as sparse or dense. Sparse retrieval refers to approaches that are based on sparse representations of the query and corpus, such as a bag of words representation (Manning et al., 2008). Sparse retrieval suffers from vocabulary mismatch problem, which is solved by dense retrieval. Dense retrieval exploits dense vector representations, or embeddings, of the queries and corpus elements and uses them to compute the similarity between query and corpus elements. Hybrid approaches combine the two by using sparse methods first to select promising candidates and then dense methods to pick from those candidates (Nogueira et al., 2019).

In dense retrieval, the vector representations of the query and the corpus elements play a very important role in determining the quality of retrieval. Models that compute these vector representations can be pretrained. Pretrained word embeddings (Mikolov et al., 2013) and sentence embeddings (Reimers & Gurevych, 2019) are widely used. The pretraining happens on unsupervised data at scale and helps create high quality vector representations of text and even code (Neelakantan et al., 2022a); for example, the OPENAI ADA embedding model is one such pretrained model.

There is renewed interest in retrievals and embeddings due to the wide adoption of generative pretrained large language models (LLMs), such as GPT-4 (OpenAI, 2023). These models have demonstrated the remarkable emergent ability of performing new tasks when provided only a few examples of the task (Brown et al., 2020a). The models' performance on new tasks also improves when they are provided with information relevant to the task in the prompt. Searching and collecting all information that may be relevant to a task is a *retrieval* problem, which leads to the retrieval-augmented generation (RAG) paradigm (Lewis et al., 2020). A popular approach to perform retrieval for RAG is to use dense retrieval; that is, find embeddings for the query and corpus and then use cosine simi-

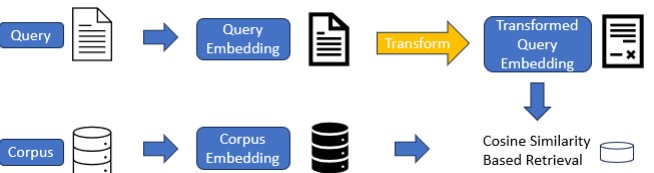

Figure 1: Overview of adapted dense retrieval: The embedding of the query is additionally transformed before doing a cosine similarity with embedding of the corpus element. In the full variant, the corpus embedding is also transformed (not shown in figure) before performing cosine similarity.

larity to pick the best artifacts for the query. However, the performance of dense retrieval for RAG applications has often been found to be lacking. This is because the retrieval problems that arise in RAG have certain features that are different from the more traditional applications of retrieval.

1. First, these retrieval problems are *heterogeneous* in nature because the corpus here may contain code, documentation, structured and semi-structured text, data, and artifacts in specialized domain-specific languages. The embeddings of these artifacts created by pretrained models (such as OPENAI ADA) may not necessarily be high quality because the pretraining data may not have had access to those kinds of special-purpose artifacts.

2. Second, the retrieval results are *strictly* required to contain all relevant artifacts in the top-$k$ for much smaller $k$. This is because the prompts sent to LLMs can include a limited amount of information (tokens) due to cost and model-induced constraints. So, here top-10 or top-50 accuracy matters much less than the top-1 or top-5 performance.

3. Finally, the retrieval problems are more *nuanced* due to task-specific notions of what is similar or relevant and what is not. This notion of similarity may not match the notion of similarity learnt by general-purpose embeddings produced by pretrained models.

The situation with pretrained embeddings is similar to the situation with pretrained language models. Pretrained generative language models (Radford & Narasimhan, 2018; Radford et al., 2019; Brown et al., 2020b) perform well on several tasks, but there are always some specialized domain-specific tasks where their performance is poor. One approach that has been used to address this issue is *fine tuning* (Wortsman et al., 2022; Liu et al., 2022). Fine tuning achieves transfer learning by further training the weights of a pretrained model on new task-specific data. In the same spirit, we ask the question: *can we fine tune pretrained embedding models with task-specific data?* This can potentially solve the three challenges mentioned above. However, fine tuning requires examples and access to weights of a model. We often have access to training data, but only in the form of a *limited number* of examples of "good" retrievals. Moreover, we often do not have access to the weights of the model, and even if we did, the cost and data requirements would be prohibitive. We solve both these issues by so-called *parameter efficient fine tuning* (PEFT) (Liu et al., 2022). We instantiate PEFT in our setting by augmenting the black-box embedding model with small "adapters". The new parameters introduced in the adapters are learnt using the limited amount of training data. The internal weights of the black-box embedding model are left untouched. The adapted embeddings are then used for retrieval. We call our approach *Adapted Dense Retrieval* or ADDER; see Figure 1. The ADDER approach is based on tuning the embedding models and this allows us to effectively support heterogeneity, strictness requirements, and semantic alignment with task-specific notions of similarity.

The adapter we attach to the black-box model in ADDER takes the form of a residual adaptation term. Intuitively, the adapter is learning the *perturbation* to be applied to the general-purpose embedding so that it becomes more task and domain specific. This is reminiscent of low-rank adaption (LoRA), which has been used to fine tune large language models by adding low rank matrices to the weight matrices of the original model (Hu et al., 2021). The low rank matrices can be viewed as perturbations. Our residual adaption is different since it does not change any weights, but rather it adds a perturbation to the embedding itself. We compute the perturbation term by passing the general-purpose embedding through a softmax based dictionary lookup. Mathematically, the dictionary lookup looks like a low rank matrix multiplication but for the fact that there is softmax layer in between the two matrices in our case.

Once the architecture for the adapter is finalized, the final step is to use the training data that contains examples of "good" retrievals to learn the adapter weights. Let us say we have a training data point where a query $q$ maps to a corpus element $c$. We train our adapter by enforcing that the adapted embedding of $q$ be closer to the adapted embedding of $c$ than to the adapted embedding of any negative corpus element $c'$. A key question here is what negative example $c'$ to use when processing the data point $(q, c)$. There are several works that describe how to pick the negative training instances for dense retrieval (Karpukhin et al., 2020). We take cue from the work (Xiong et al., 2020) that suggested taking hard negatives from the full corpus. Specifically, instead of random or in-batch local negatives, we find global negatives using the current state of the adapted model on the entire corpus.

## 2  HETEROGENEOUS STRICT RETRIEVAL

Retrieval is being increasingly used to build applications that are based on pretrained large language models (LLMs). Embeddings from LLMs Neelakantan et al. (2022b); Touvron et al. (2023) have been used extensively to support the retrieval component in such applications. However, some of these new applications of retrieval pose different set of challenges from what were posed by other more traditional uses of retrieval. Motivated by the new applications, we introduce the class of *heterogeneous strict retrieval* problems where we are given three inputs:

- A single query $q_0$ from a domain $\mathbb{Q}$ of queries;
- A corpus $\mathbb{C}$ of candidate items;
- A dataset $\mathbb{D}$ containing pairs $(q, c)$, where $q \in \mathbb{Q}$ and $c \in \mathbb{C}$.

The goal is to extract some $k > 0$ elements from $\mathbb{C}$ that are most likely to be related to the query $q_0$ in the same way that pairs in $\mathbb{D}$ are related. The focus of our work is on cases where $k$ is required to be small. We use the term *strict* retrieval to emphasize the fact that we want the most likely candidate to appear in the top-1 or top-3 or top-5 of the retrieved candidates (as opposed to being in the top-10 or top-50).

In the applications we are targeting, the corpus $\mathbb{C}$ typically contains a few hundreds or a few thousands of elements. Similarly, the dataset $\mathbb{D}$ contains a few hundreds or thousands of $(q, c)$ pairs. A key distinction with most earlier work on retrieval is that we are targeting *heterogeneous* retrievals where the domains $\mathbb{Q}$ and $\mathbb{C}$ are potentially very distinct. For example, the artifacts in $\mathbb{Q}$ may be natural language sentences, whereas those in $\mathbb{C}$ may be code snippets, or semi-structured or structured text or data.

We assume that we have black-box access to a pretrained large language model that can compute embeddings. Let $E : (\mathbb{Q} \cup \mathbb{C}) \mapsto \mathbb{L}$ be the black-box function that given an entity – either a query $q$ from $\mathbb{Q}$ or a corpus element $c$ from $\mathbb{C}$ – computes its representation $E(q)$ or $E(c)$, respectively, in some (common) latent representation space $\mathbb{L}$. For our experiments, we will use the text embedding model OPENAI ADA as the function $E$. We can use the model to compute (general-purpose) embeddings, but we assume we have no ability to fine-tune or refine the model in any way. This assumption is reasonable since access to model weights is often restricted, and furthermore, fine-tuning the models can be cost and time prohibitive. Moreover, we typically have only small amounts of data in set $\mathbb{D}$ that may not be sufficient for fine tuning.

### DENSE RETRIEVAL

We first recall dense retrieval, which serves as a baseline for solving our *heterogeneous strict retrieval* problem.

The embeddings computed by pretrained large models are *dense* (as opposed to sparse), and such embeddings can be used for retrieval using the following function, $\texttt{ret}_{\texttt{oai}}(k, q_0, \mathbb{C})$, which retrieves $k$ candidates from the corpus $\mathbb{C}$ for a given query $q_0$:

$$\texttt{ret}_{\texttt{oai}}(k, q_0, \mathbb{C}) \quad ::= \quad \underset{c \in \mathbb{C}}{\arg\max\texttt{K}} \ \ \texttt{sim}(E(q_0), E(c)) \tag{1}$$

where $\texttt{sim}$ is a measure of similarity between two vectors in the embedding space $\mathbb{L}$ (Xiong et al., 2020; Lee et al., 2019; Luan et al., 2021; Karpukhin et al., 2020). We use cosine similarity as the

measure in this work; thus, the $\arg\max\text{K}$ operator is simply returning the $k$ corpus elements that are the $k$ nearest neighbors of $E(q_0)$ in the latent space $\mathbb{L}$.

The embeddings computed by $E$ are general-purpose and can be used as latent representations for various entities, such as, code and natural language. The function $\text{ret}_{\text{oai}}$ defined in Equation 1 provides a very strong baseline for retrieval, even for *heterogeneous strict retrieval*, since large language models have been trained on vast amounts of data. However, it relies on a fixed notion of similarity, which is not informed by the specific notion of similarity that may be of interest. In our setting, we are given the set $\mathbb{D}$ of pairs of query and corpus elements that illustrates the notion of similarity that is desired. If the similarity notion implicit in $\mathbb{D}$ matches the one induced by $E$ and $\text{sim}$, then dense retrieval, as performed by $\text{ret}_{\text{oai}}$, would perform just as well. However, if there is a mismatch, then we need to *adapt* the embedding so that it captures the semantics used in the relation between the example pairs in $\mathbb{D}$.

**Remark.** Note that we are using the same embedding calculator $E$ for both query and corpus in Equation 1 only for simplicity of presentation. We can generalize easily to the case when we have different functions, say $E$ and $E'$, to embed queries and corpus elements respectively as long as the target latent space $\mathbb{L}$ is shared. All subsequent developments also generalize naturally.

## 3 ADAPTED DENSE RETRIEVAL

Our approach for addressing the *heterogeneous strict retrieval* challenge is based on adapting the black-box embedding $E$ to the specific notion of similarity presented in the dataset $\mathbb{D}$. We call our approach *adapted dense retrieval*.

We adapt the embedding function $E$ by simply composing it with another function $\text{Tr} : \mathbb{L} \mapsto \mathbb{L}$ so that the initial query $q_0$ is now transformed into $\text{Tr}(E(q_0))$. The transformation $\text{Tr}$ is learnt from the dataset $\mathbb{D}$, which we will describe later. Thus, the dense retrieval function $\text{ret}_{\text{oai}}$ from Equation 1 is replaced now by the following *adapted dense retrival*, or ADDER, function $\text{ret}_{\text{adr}}(k, q_0, \mathbb{C})$:

$$\text{ret}_{\text{adr}}(k, q_0, \mathbb{C}) \quad ::= \quad \underset{c \in \mathbb{C}}{\arg\max\text{K}} \quad \text{sim}(\text{Tr}(E(q_0)), E(c)) \tag{2}$$

The ADDER function is illustrated in Figure 1. The reader may notice the asymmetry in Equation 2 wherein the transformation $\text{Tr}$ is applied only the embedding of the query. We can also apply another transformation $\text{Tr}'$ on the corpus element to get a variant $\text{ret}_{\text{adr2}}(k, q_0, \mathbb{C})$ of adapted dense retriever that we call an ADDER2:

$$\text{ret}_{\text{adr2}}(k, q_0, \mathbb{C}) \quad ::= \quad \underset{c \in \mathbb{C}}{\arg\max\text{K}} \quad \text{sim}(\text{Tr}(E(q_0)), \text{Tr}'(E(c))) \tag{3}$$

Thus, our ADDER approach retrieves $k$ nearest neighbors of the transformed embedding of the query. The purpose of the transformations $\text{Tr}$ and $\text{Tr}'$ is to capture any task-specific adaptations required on the embedding before it is used for nearest neighbor search.

The overall approach of adapted dense retrieval can work with transformations $\text{Tr}$ and $\text{Tr}'$ taken from any rich class of transformations. In this paper, we start with very simple classes of transformation for picking $\text{Tr}$ and $\text{Tr}'$, which is described next.

### 3.1 KEY-VALUE LOOKUP BASED RESIDUAL ADAPTATIONS

Inspired by prior work on low-rank adaptations that were used for fine-tuning large language models (Hu et al., 2021), we use residual adaptation as the transformation function $\text{Tr}$ and $\text{Tr}'$. Specifically, the two transformations add a residual term to the embedding $e$ computed by the black-box model; that is,

$$\begin{aligned}
\text{Tr}(e) \quad &::= \quad e + f(e, \theta) \\
\text{Tr}'(e) \quad &::= \quad e + f'(e, \theta')
\end{aligned} \tag{4}$$

The functions $f$ and $f'$ are parameterized by $\theta$ and $\theta'$ respectively that are learned from data. The neural architecture for computing $f$ and $f'$ is inspired by key-value dictionary lookup in an attention mechanism.

Since $f'$ is handled the same way as $f$, let us just focus on $f$. The function $f$ is computed by extrapolating from a table that contains mapping from keys (inputs of $f$) to values (outputs of $f$). Concretely, let $d$ be the dimension of the latent space $\mathbb{L}$ where the embeddings computed by the black-box model reside. For example, for OpenAI ada embeddings, $d$ is 1536. Let $h < d$ be a small number compared to $d$. The typical values we use for $h$ range from 16 to 128. Let $K$ be a $(h \times d)$-matrix of keys (containing $h$ keys), and let $V$ be a $(h \times d)$-matrix of values (containing $h$ values). The matrices $K, V$ define the unknown parameters $\theta$, and hence given the input embedding $e$ (of shape $(1 \times d)$), the function $f$ is then defined as:

$$f(e, K, V) \quad := \quad \texttt{softmax}(eK^T)V \tag{5}$$

where $K^T$ is the transpose of $K$. Note that since $h < d$, even though the final residual term is $d$ dimensional, all residuals technically lie in some smaller $h$ dimensional space.

Our low-dimensional residuals are reminiscent of low-rank adaption from Hu et al. (2021). However, unlike that work, our approach treats the model as a black-box and we do not adapt the weights of the model.

We have thus parameterized $f$ (and $f'$) by $2dh$ unknowns, and we next describe the loss function that can help us learn the unknown parameters from the given dataset $\mathbb{D}$ containing pairs $(q, c)$ of query $q$ and corpus element $c$.

## 3.2 GLOBAL NEGATIVES CONTRASTIVE LOSS

Finally, we need to learn the parameters $K, V$, or $\theta$ in short, for the transformation $\texttt{Tr}$ to obtain our adaptive dense retrievers outlined in Equation 2 and additionally parameters $K', V'$, or $\theta'$ in short, for $\texttt{Tr}'$ to obtain the retriever in Equation 3.

The dataset $\mathbb{D}$ contains pairs $(q, c)$, so for each $q$ in the dataset, we have examples of corpus elements $c$ that should be retrieved. Every other corpus element becomes a negative example; that is, for query $q$, the positive corpus elements $\mathbb{C}^{q+}$ and the negative elements $\mathbb{C}^{q-}$ are defined as follows:

$$\mathbb{C}^{q+} \quad ::= \quad \{c \in \mathbb{C} \mid (q, c) \in \mathbb{D}\} \tag{6}$$
$$\mathbb{C}^{q-} \quad ::= \quad \mathbb{C} \setminus \mathbb{C}^{q+} \tag{7}$$

Let $g(q, c)$ be the following function that is used in our adapted retriever and that is computed by our neural network:

$$g(q, c) \quad ::= \quad \texttt{sim}(\texttt{Tr}(E(q)), \texttt{Tr}'(E(c))) \tag{8}$$

Learning to retrieve is the same as learning to rank (Liu, 2009), and hence, to learn the best possible $g$ for retrieval, we need to minimize the loss over all pairs of positive and negative samples for a query; that is,

$$\theta^*, {\theta'}^* \quad ::= \quad \underset{\theta, \theta'}{\arg\min} \sum_{(q,c) \in \mathbb{D}, c^- \in \mathbb{C}^{q-}} l(\texttt{Tr}(E(q)), \texttt{Tr}'(E(c)), \texttt{Tr}'(E(c^-))) \tag{9}$$

If $e := \texttt{Tr}(E(q))$, $e^+ := \texttt{Tr}'(E(c))$, and $e^- := \texttt{Tr}'(E(c^-))$, then the loss function $l(e, e^+, e^-)$ should be zero if $e$ is closer to $e^+$ than it is to $e^-$ and it should be positive otherwise. Since the set $\mathbb{C}^{q-}$ can be very large, often one approximates by sampling from this set and using the sample in Equation 11 in place of $\mathbb{C}^{q-}$. Typically, the sample is restricted to the corpus elements that appear in the batch being processed. However, recent work showed that this leads to poor learning (Xiong et al., 2020). That work also emphasized the need for finding negative samples from the full corpus. We observed the same behavior, and hence we picked the negative corpus element that was closest to the query embedding as *the* negative sample to use in Equation 11. In other words, we use the following modification of Equation 11:

$$\theta^*, {\theta'}^* \quad ::= \quad \underset{\theta, \theta'}{\arg\min} \sum_{(q,c) \in \mathbb{D}} l(\texttt{Tr}(E(q)), \texttt{Tr}'(E(c)), \texttt{Tr}'(E(\texttt{GlobalNeg}(q)))) \tag{10}$$

where the function $\texttt{GlobalNeg}$ is computed as follows:

$$\texttt{GlobalNeg}(q) \quad ::= \quad \underset{c \in \mathbb{C}^{q-}}{\arg\max} \ \texttt{sim}(\texttt{Tr}(E(q)), \texttt{Tr}'(E(c))) \tag{11}$$

Note that the ideal negative sample, $\texttt{GlobalNeg}(q)$, for query $q$ depends on the functions $\texttt{Tr}$ and $\texttt{Tr}'$ that we are learning. Thus, $\texttt{GlobalNeg}(q)$ has to be calculated during training in every batch. This is not an issue since we have relatively small training datasets and modest corpus sizes.

Table 1: Metadata for BEIR benchmarks used for evaluation.

| Name | Partitions | Queries Used | Corpus size |
|------|------------|--------------|-------------|
| SciFact | train, test | 919 train, 339 test | 5K |
| ArguAna | test | 1406 test split into 1124 train,282 test | 8.67K |
| FiQA | train, dev, test | 14166 train, 1706 test | 57K |
| NFCorpus | train, dev, test | 11385 dev as train, 12334 test | 3.6K |

Table 2: Retrieval Performance on BEIR Datasets. The metrics for SBert were uniformly worse than those for OpenAI embeddings, and hence they are not reported here. Boldface denotes a win for the tool in that category and underline denotes a tie.

| Benchmark | OAI nDCG@ | | | | ADDER nDCG@ | | | | ADDER2 nDCG@ | | | |
|-----------|------|------|------|------|------|------|------|------|------|------|------|------|
| | 1 | 3 | 5 | 10 | 1 | 3 | 5 | 10 | 1 | 3 | 5 | 10 |
| SciFact | 0.62 | 0.69 | 0.70 | 0.73 | 0.65 | 0.71 | 0.77 | 0.77 | **0.74** | **0.80** | **0.81** | **0.83** |
| ArguAna | 0.32 | 0.49 | **0.54** | **0.59** | **0.37** | **0.49** | 0.53 | 0.57 | 0.36 | 0.48 | 0.52 | 0.56 |
| FiQA | **0.41** | **0.39** | **0.40** | **0.43** | 0.40 | 0.37 | 0.39 | 0.41 | 0.39 | 0.37 | 0.38 | 0.41 |
| NFCorpus | 0.46 | 0.42 | 0.40 | 0.37 | **0.47** | **0.43** | 0.40 | 0.37 | 0.45 | 0.40 | 0.37 | 0.35 |

## 4 EXPERIMENTAL EVALUATION/RESULTS

We next experimentally evaluate *adapted dense retrievals* in different ways. The goal is to determine if adaptation adds any value over off-the-shelf OPENAI ADA embeddings.

When we use the approach from Equation 2, we refer to the system as ADDER. When we use Equation 3, the system is called ADDER2. The main baseline is OAI, which denotes the use of standard OPENAI ADA embeddings (*text-embedding-ada-002*[1]). We carried out our experiments on regular laptops and desktops, and used no special purpose hardware for training or inference except for the black-box rest API calls to the OpenAI embedding endpoint.

*Experimental Setup:* To be precise, for our experiments over ADDER & ADDER2, we use a virtual machine with a single Nvidia K80 GPU (with 24GiB of vRAM), 4 CPU cores and 28 GB of RAM. The optimization is done using the Adam optimizer (Kingma & Ba, 2014) for both mechanisms. For training of ADDER, we start an initial learning rate of $1e^{-3}$ with a step update of $\gamma = 0.5$ every 100 epochs. On the other hand, for ADDER2, we start with a learning rate of $1e^{-2}$ and a step update with $\gamma = 0.5$ every 50 epochs. The training for all cases was terminated after 500 epochs (although learning saturated in most cases well within the terminal limit). The best model weights are determined using hyperparameter tuning in a range of hidden layer dimensions between 64 and 1536.

### 4.1 ADDER IMPROVES RETRIEVAL ON CERTAIN IR BENCHMARKS

We first evaluate adapted dense retrieval on classical retrieval benchmarks. We use benchmarks from the BEIR collection (Thakur et al., 2021). In particular, we focus on four datasets from the BEIR collection: SciFact, ArguAna, FiQA and NFCorpus. These were the four smallest sets in the BEIR collection, see Table 1, and thus they aligned with our motivation of focusing on tasks where there was limited data. They are also a reasonably diverse set of heterogeneous retrieval use cases. The dev test was used to train the model in case of NFCorpus. ArguAna contains only a test set, so we did a 80-20 split to create train-test partitions. OAI needs no training for retrieval, but ADDER and ADDER2 were trained on the particular training sets, and all were evaluated on the test sets.

Table 2 presents the standard "normalized discounted cumulative gain" (nDCG) values at $k = 1, k = 3, k = 5$, and $k = 10$ for the 4 benchmarks and the 3 systems. SciFact and ArguAna datasets associate only one corpus element with any query. Hence, for these two cases, nDCG@$k$ can only increase with $k$. That is not true for the FiQA and NFCorpus.

---

[1]https://openai.com/blog/new-and-improved-embedding-model

Table 3: NL2X Retrieval Metrics

| | OAI nDCG@ | | | | ADDER nDCG@ | | | | ADDER2 nDCG@ | | | |
|---|---|---|---|---|---|---|---|---|---|---|---|---|
| X | 1 | 3 | 5 | 10 | 1 | 3 | 5 | 10 | 1 | 3 | 5 | 10 |
| SMCALFLOW | 0.64 | 0.72 | 0.74 | 0.75 | **0.96** | **0.98** | **0.98** | 0.98 | 0.95 | 0.97 | 0.97 | 0.98 |
| BASH | 0.75 | 0.84 | 0.85 | 0.86 | 0.78 | 0.88 | 0.89 | 0.89 | **0.79** | 0.88 | 0.89 | 0.89 |
| CONALA | **0.77** | **0.87** | **0.88** | 0.88 | 0.75 | 0.85 | 0.86 | 0.86 | 0.75 | 0.86 | 0.87 | 0.88 |

The systems ADDER and ADDER2 both perform better than OAI on SciFact and ArguAna in terms of the nDCG metrics @1, 3, 5. For FiQA and NFCorpus, the adapted retrivers, ADDER and ADDER2, are unable to add much value over OAI and mostly perform almost as good as OAI.

The results show that adapted dense retrieval can improve performance of black-box pretrained embedding models by learning a transformation that is customized to the particular retrieval task. This tuning of the pretrained embedding does not need huge amounts of training data. Task-specific adaptation helps when the train set contains enough examples of retrieving the various artifacts in the corpus. In cases where it does not improve performance metrics, adaptation does not make the metrics much worse and performance remains almost at par with the black-box embedding model. We hypothesize that the differences in gains achieved by ADDER and ADDER2 across benchmarks is due to the differences in alignment of the notion of semantic similarity learnt by OPENAI ADA with what is intended in the benchmarks. For example, SciFact uses very stylized queries and the notion of similarity is that the query is either supported by or refuted by the corpus text. The adapter is possibly able to tweak the embeddings to match the benchmark intention for this set. The notion of similarity is closer to NL semantic similarity for NFCorpus, so there is little opportunity to gain by adaptation.

## 4.2 ADDER SIGNIFICANTLY IMPROVES NL2X RETRIEVALS

We next evaluate the adapted dense retrieval on heterogeneous tasks where the query is in natural language, but the corpus elements are code fragments. Our hypothesis is that adaptation will help more in these cases since the query and corpus elements are different kinds of entities. While the BEIR benchmarks were also heterogeneous, the query and corpus were both natural language expressions. A second interesting goal here is to study the impact of the popularity of the programming language on the performance of baseline embeddings and the adapted embeddings.

We picked three NL2X datasets from the public domain. The three target languages X we picked were SMCALFLOW, BASH, and PYTHON. These language cover the spectrum from low resource, special-purpose, and less-known language, SMCALFLOW, on one end to popular, general-purpose and widely-used language, PYTHON, on the other end with BASH somewhere in the middle. The NL2SMCALFLOW benchmark(Semantic Machines et al., 2020; Platanios et al., 2021) consists of user utterances (in natural language) about tasks involving calendars, weather, places, or people, which is paired with an executable dataflow program written in lisp-like syntax. The NL2BASH benchmark (Lin et al., 2018) consists of natural description of a task paired with a bash command that accomplishes that task. The NL2PYTHON benchmark Yin et al. (2018) comes from data collected from StackOverflow, and contains pairs of natural language and aligned code (in PYTHON). Each benchmark has a well-defined train and test split provided.

The three NL2X benchmarks were originally created for the task of generating code from natural language. Here, we repurpose them for retrieval: given the natural language utterance and the corpus of program expressions, retrieve the code corresponding to the NL utterance. We remark here that this retrieval task is motivated by few-shot selection. In code generation from NL tasks, one has to pick those NL-Code pairs as few-shots where the code is most similar to the code expected from the NL query. Any progress on our repurposed task will facilitate few-shot selection.

As before, we can solve this retrieval task using OPENAI ADA embeddings (OAI), or we could use our approach and adapt the embeddings using the train set and then use them for retrieval (ADDER and ADDER2). Table 3 presents the results of retrieving code from the test set given the corresponding NL utternace from the test set. For NL2SMCALFLOW, ADDER and ADDER2 both give a huge gain over OAI in the nDCG@$k$ metric for all values of $k$. The gains are still there for NL2BASH

benchmark, but they are a bit muted. The gains disappear for NL2PYTHON (CONALA), and OAI starts out-performing our models, but only by a slight margin, for small values of $k$.

There are a few reasons why OPENAI ADA embeddings do well on the CONALA dataset. The first is that the target language is PYTHON, and the pretrained embedding models are known to be fairly good at Python. The second is that the NL descriptions of the PYTHON code are of very high quality. The CONALA dataset contains "cleaned up" versions of the original NL annotation extracted from STACKOVERFLOW. These NL descriptions were fixed by humans whenever they were found to not describe the code accurately. We used these "fixed" versions in our experiments. Since the NL describes the code fairly accurately, so its embedding (computed by OPENAI ADA embeddings) is naturally very close to the embedding of the associated PYTHON code, and hence there is not much scope for our technique to improve it. Finally, and most importantly, we realized after the experiments were completed that the mapping from NL to PYTHON code in the CONALA dataset is one-to-many. While it is true that our technique can work with one-to-many relations, it does so only after we have prepared the dataset by collecting all possible corpus elements that correspond to the same query, and since we did not perform this preprocessing, our approach struggles on the CONALA dataset.

In contrast to PYTHON, SMCALFLOW is an unfamiliar target and the OPENAI ADA model does not do as well on embedding those programs. Furthermore, the train-test split in the SMCALFLOW benchmark is such that the train set contains good representatives for the test queries, and hence, our adaptation performs extremely well on it.

## 5 RELATED WORK

End-to-end Information Retrieval (IR) systems colloquially have four major components (Zhu et al., 2023), namely (a) query rewriter, (b) retriever, (c) re-ranker and (d) reader. Our proposed technique ADDER potentially impacts the first three components, either directly or indirectly, by abstracting out the input representation and directly tuning the meta-representations (embeddings). Embeddings are at the core Neural IR. Particularly, recent advances in language understanding demonstrate how off-the-shelf LLMs and their embedding representations (Neelakantan et al., 2022b; Touvron et al., 2023) are being leveraged in both upstream and downstream components of the IR pipeline (Dodge et al., 2020; Asai et al., 2019; Khashabi et al., 2020). This motivates our works on improving retrieval and its applications with minimal augmentations to off-the-shelf embeddings.

### QUERY REWRITE

In the IR domain, query formulation and expansion have been considered as essential steps to improve upstream retrieval. Classic techniques, such as pseudo-relevance feedback (PRF) (Salton & Buckley, 1990) and Rocchio's relevance feedback (Rocchio Jr, 1971), have been used to refine initial queries based on feedback from relevant documents. Moreover, external sources, such as thesauri and ontologies, have been employed to expand queries (Carpineto & Romano, 2012). These methods have primarily operated in the textual space by manipulating query terms, adding synonyms or additional metadata to improve the robustness of the retrieval.

Our work potentially achieves the goals of query expansion, but using a significantly different approach. Rather than work on the input (textual) domain, we focus on the embedding space. We learn adapters that automatically perform query embedding refinement (Carpineto & Romano, 2012). The refined embedding can be viewed as an embedding of the (unknown) rewritten query. Our approach leverages the inherent structure and semantics captured in embeddings, and thus allows us to adapt queries in a data-driven, task-specific fashion, making our approach particularly effective in scenarios where queries and corpus elements involve diverse types of entities, such as NL-code and query-arguments, with a strict retrieval policy requirement.

### RETRIEVER

The retrieval component of the pipeline varies widely according to the downstream task associated with the system. Here, the query and the search space could be either homogeneous (Voorhees, 1999), that is, the same information modalities, such as NL-NL, or heterogeneous (Thakur et al., 2021), that is, different but dependent information modalities, such as NL-Code. While building

retrieval systems for dissimilar model types has always been a challenging problem, the use cases of such retrievals have been gathering momentum due to proliferation of LLM-based tools. Various approaches have been proposed for heterogenous retrieval, including learning embeddings in a joint cross-modal fashion by sharing semantic information in a common representational space within the vision-language community for cross-modal alignment and recommendation systems (Ma et al., 2021).

Joint Embedding Predictive Architectures (JEPAs) (Assran et al., 2023) extend this idea further by not only mapping data into a shared representational space, but also learning predictive functions that can identify the compatibility or similarity between inputs. This predictive capability, besides being useful to act over user's history to present recommendations, can also be used to learn and suggest entities compatible with their preferences (Assran et al., 2023). Our technique draws inspiration from such architectures where we train ADDER with the loss function in the (possible) shared latent embedding space for heterogeneous entity pairs and optimizing the similarity in this representation. This space is tuned to emphasize the aspects of the representation that are learned from the data for a given downstream task. It can then be used as the ground for making multiple observations about the closeness between these dissimilar entities.

### RERANKER

The Reranker aims to reorder the retrieved entities so that the top entities are more aligned with the query and task at hand. It involves sophisticated methods of ranking that consider more subtle features of the retrieved entities, such as term-proximity, quality, and relevance feedback (Lv & Zhai, 2009; Valcarce et al., 2018; Wang & Zhu, 2009). Neural models have been increasingly employed in learning to re-rank for a particular task setting (Cao et al., 2007; Thakur et al., 2021; Cormack et al., 2011; Dumais et al., 2003; Huang & Hu, 2009). ADDER enables developers to try out various types and combinations of loss functions, and tune the adaptation to align with their preferred ranking suited to their tasks. We do this with modest data pre-requisites and without learning complicated models that represent the ranking space.

### READER

In the context of the emerging paradigm of Retrieval Augmented Generation (RAG) (Lewis et al., 2020), the Reader component plays a crucial role by synthesizing relevant information from retrieved information entities. RAG combines traditional information retrieval with natural language generation, often involving large language models, such as GPT (Radford & Narasimhan, 2018; Radford et al., 2019; Brown et al., 2020b; OpenAI, 2023), BERT (Devlin et al., 2018; Akkalyoncu Yilmaz et al., 2019a; Trabelsi et al., 2021) and T5 (Raffel et al., 2019); some of which are tuned on retrieval tasks (Chung et al., 2022; Asai et al., 2022). While ADDER does not directly help with the goals of the reader component, it can potentially empower these newer applications, such as RAG and few-shot selection, through its role in the other three retrieval components.

## 6 CONCLUSION

We presented adapted dense retrieval – a dense retrieval approach that additionally adapts the embedding and aligns it with any task-specific notion of similarity or closeness. The specific adapter we consider in this paper adds a residual term to the general-purpose embedding. We considered a very simple adapter that maintains a key-value dictionary and computes the residual by softmax-based lookup in this dictionary. The benefit of using a small adapter is that it can be trained using only a limited amount of annotated task-specific data. In future work, we plan to consider richer architectures for the adapter. We experimented with two kinds of adapter architectures: one where we adapt only the query embedding and the other where we adapt both the query and corpus embeddings.

Just as fine tuning of generative models is only needed in special cases, adapters for embeddings are necessary only when the default embeddings are of poor quality for the entities of interest. This usually happens when the query or corpus elements are uncommon entities that are unfamiliar to pretrained models, such as code in a niche domain-specific language.

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
