# OpenReview forum: "Adder: Adapted Dense Retrieval"
_ICLR.cc/2024/Conference — Submitted to ICLR 2024_

### Official Review · Reviewer_YJBY · 2023-10-30

**Soundness:** 2 fair
**Presentation:** 3 good
**Contribution:** 1 poor
**Rating:** 3
**Confidence:** 4

**Summary:**

- Authors propose to a transformation function on top of embeddings from a fixed model/API to adapt them to a task-specific notion of similarity.
- The transformation function is learned using task-specific labelled data that contains positive (query, item) pairs marked as relevant, and hard negative mining strategies are used for finding negatives items while training the transformation function.

**Strengths:**

- The paper is fairly well-written and easy to follow.
- Well-motivated: I agree with the motivation for learning a small set of domain-specific parameter to learn some domain specific notion of similarity.

**Weaknesses:**

- Limited novelty and limited empirical results
  - The proposed approach has limited novelty and is similar to existing work of training task/domain specific parameters while keeping model parameters fixed. In this case, the model is actually served through an API and users can only access final embedding outputs from the model.
  - This proposed approach is similar to adding additional trainable parameters on top of final layer of a model. Even the choice of transformation function used in this paper is explored in prior work.
  - In terms of the transformation function, only on formulation is tried. And no ablation/alternates are tried.
     - There can be other simple variations such as using a shallow multi-layer perceptron model, using trainable skip connection weight parameter etc. Adding such other choices can better further strengthen experiments in this paper.

- Evaluation only on small-scale dataset and limited analysis:
   -  The domains used for evaluation are small-scale containing up to 57K items. This is rather small scale for information retrieval.
   -  Why are large scale datasets from BeIR benchmark not used? It contains domains of the size of up to 8 million items.
   - Even for these small-scale datasets, adding analysis such as effect of varying training data size, understanding effect of corpus size, effect of using pseudo-queries (released as part of BeIR benchmark) vs using actual train queries etc can further strengthen the paper.

**Questions:**

Some minor suggestion for writing:
- I would suggest using \mathcal{C} instead of \mathbb{C} for denoting corpus. (\mathbb{C} is typically used for complex numbers)
- I would use \paragraph{} instead of \subsection{*} for different subsections/paras in related work.

---

> ### Author Response · Authors · 2023-11-17
> **Response for review by Reviewer YjBY**
>
> The review correctly points out that our approach uses some well-known ideas, like residuals and adapters. The key novelty is the combination of these ideas in the right way and applying them on embeddings. We are thus able to tune retrievals to solve a problem that the review agrees is well motivated. In short, the contributions include the strict heterogeneous retrieval problem definition that arises from new RAG-like applications, the idea of adapting embeddings, the idea of using residual low-rank adaptations for embeddings and the evaluation showing that the idea works.
>
> The review also correctly points out that evaluation is limited in the paper. We performed extensive experiments with various modeling choices, such as not using residuals, not using contrastive loss, not using global hard negative, etc., before we arrived at the right solution presented in this paper. Unfortunately, we did not include those results in the paper. We plan to fix this oversight and include the results for these variants: (1) not using residual adaptation and instead using direct MLP adapter, (2) not using global hard negatives and instead using batch negatives, (3) not using contrastive loss. We also experimentally observed that Adder and Adder2 adapters over SBert embeddings show an even greater relative gain in nDCG metrics than what we report for OpenAI embeddings. Please see the table in the response above to Reviewer Y6UA.
>
> Regarding the limited corpus size, the choice is guided purely by our end use case for RAG.  Also in our target application, we have limited training data. We wanted to design a solution that can work with limited training data, even if it required small corpus sizes, and used limited computational resources. That was the goal of our work. We agree that for larger scale datasets and enough training data, other approaches, such as finetuning, could come into play, but they would also incur a significant cost and time. ADDER only takes 8secs for 1 epoch (668 batches for a batch size 16) on a single gpu (k-80) machine.

---

> > ### Comment · Reviewer_YJBY · 2023-11-19
> >
> > Thank you for providing the clarifications.
> > I believe that the paper will benefit from adding more experiments on larger domains as the current domains seem to be small for real applications. In terms of lack of training data for larger domains, authors can maybe use pseudo-queries released as part of BeIR benchmark for running experiments on domains without real queries.
> >
> > I think main strength of this paper can be a thorough empirical investigation as the ideas presented are simple and have limited novelty. And my reasons to suggest that this paper is not ready for the conference yet are not that it is simple but that the current experiments are somewhat on all small datasets with limited analysis and ablations.

---

> > > ### Author Response · Authors · 2023-11-23
> > >
> > > We thank the reviewer for helping us with suggestions for improving the paper.
> > > However, the comment about "lack of training data" made us think that the reviewer might have misinterpreted our original response.
> > >
> > > We would like to clarify that the training data is limited by design in our proposed technique & experiments to cater to low-resource domains (e.g. downstream applications of RAG) where there is limited access to data. Our whole approach and the design choices -- adapters and hard global negatives -- are guided by this limited data setting. Hence, the evaluation is also focused on the scenarios where data is limited.
> > >
> > > The immediate end goal is not to perform general-purpose information retrieval, but a very specific application of it needed for LLM-based solutions for tasks such as NL2Code; please also see our overall response at https://openreview.net/forum?id=n3kFlvVhJM&noteId=UiCjIw5BCv.  We recently obtained experimental results that show the gains from using ADDER and ADDER2 for NL2Code tasks performed using LLMs where we (a) use ADDER to retrieve few-shot examples and (b) use ADDER to rerank the candidates returned by the LLM. This application was one of our main motivations for designing ADDER and ADDER2. On the Excel's PowerQuery Mashup dataset, we observe that using only OpenAI embeddings gives us a sketch match performance of 0.72, whereas using ADDER gives us a performance of 0.74.  The important point to note here is that the Mashup is a low-resource language with limited data availability, thus it is not amenable to techniques for domain-adaptation that require high data requirement.
> > >
> > > We would also like to point the author to some qualitative examples for the same task before & after we apply ADDER to the pipeline in the [official comment on the main thread](https://openreview.net/forum?id=n3kFlvVhJM&noteId=HHbXMvYBvT). The reviewer may note the difference in quality for the top-ranked examples.
> > >
> > > If the training data was not a constraint, many existing approaches would be available -- finetune/retrain white-box models, training dense layers on top of the black-box model --  which could potentially perform better. ADDER employs residual adaptation to embeddings; this allows us to massage the inherent distribution of data to adapt better to the task but not transform it completely. Over larger datasets, we suspect that transformation of the embeddings would be of a much higher degree and denser to learn, but that is not the goal here.
> > >
> > > We also reiterate that the paper is more than just an empirical evaluation. There is a novel technical contribution here, which uses well-known pieces to stitch together a solution for a very specific, and important, problem. Even though we used BEIR benchmarks for evaluation, general-purpose IR is not our main goal, but some of the selected BEIR benchmarks do fit our target goal and hence our evaluation on the selected subset.

---

> > > > ### Author Response · Authors · 2023-11-23
> > > >
> > > > We are reminding all reviewers to also see the consolidated responses at https://openreview.net/forum?id=n3kFlvVhJM&noteId=UiCjIw5BCv and https://openreview.net/forum?id=n3kFlvVhJM&noteId=HHbXMvYBvT (apart from the individual responses), if they have not already done so.

---

### Official Review · Reviewer_Y6UA · 2023-11-01

**Soundness:** 3 good
**Presentation:** 3 good
**Contribution:** 2 fair
**Rating:** 6
**Confidence:** 4

**Summary:**

This paper proposes a new task-adaptation method for dense retrieval. It takes off-the-shelf embeddings and learns a one-layer residual adapter using task-specific training data. This adapter will transform the embedding into a more task-specific one. The authors evaluated the proposed method on 4 text retrieval tasks from BEIR and 3 code retrieval tasks derived from NL2X.  Experimental results show that the adaptation method can improve retrieval quality for some tasks, especially at very high rank region (1 to 5).

**Strengths:**

- Paper is clearly written and well presented.
- Though there are many existing work on task-adaption of retrievers, most of them change model weights. This paper treats the model as a black-box and directly transforms the embeddings, which is less-explored but very practicle.
- Experiments show good improvements on some tasks.

**Weaknesses:**

- The modeling choices need to be better justified. I wonder how a simple MLP adapter layer performs comparing to the proposed attention + residual approach.
- The paper can be stronger with more ablations and analysis. One important experiment is to test if the proposed method can work with different base embeddings models, not just OpenAI embeddings. Another interesting ablation is the global hard negative, as it brings much technical complexity. In addition, it would be nice to show the quality with various amount of training data.
- Would be nice to report results on the rest of the BEIR datasets.
- Section 4.2 is vague. Training and testing data size and training set up are missing.

**Questions:**

- Why using the attention + residual adaptation? Have you tried other options, e.g., a linear adapter or MLP?
- How important is the global hard negatives? What are the implementation details, e.g., do you refresh the index at every training step?

---

> ### Author Response · Authors · 2023-11-17
> **Response to review by Y6UA**
>
> Q1: It is a good question about why attention + residual. Indeed, we started with a linear adapter and an MLP adapter. We observed that their performance was often much worse than the raw openAI embeddings, especially when there was limited training data. We realized that OpenAI embeddings are actually very good, and we wanted to keep their “goodness” and only perturb/tweak them slightly for a particular retrieval task. If the reviewer would find it useful, we can rerun those early experiments and report the results.
>
>  Q2: The story is the same for global hard negatives. We started with not using contrastive loss at all, then we moved to batch negatives, and finally to global negatives because we saw improvements. For Adder2, where we also adapt the embedding of the corpus, we refresh corpus embeddings at each epoch.  In Adder, we do not adapt the corpus embeddings, so this is not an issue.
>
> W1, W2: We realize now that although we did a lot of experiments with other modeling choices, we only reported the numbers with the best architecture we found. We will include performance numbers for following ablations in the final version: (1) not using residual adaptation and instead using direct MLP adapter, (2) not using global hard negatives and instead using batch negatives, (3) not using contrastive loss.
>
> Regarding using other base embeddings, thank you for the suggestion because we ran experiments with sBert and noticed that the relative gains provided by Adder are huge (compared to the relative gains we get over OpenAI embeddings). Although, the absolute numbers are poorer than what we get with OpenAI embeddings.
>
> Here are the results:
> For the same BEIR benchmarks reported in Table 2, here are the results with using SBert as base. (nDCG@k values)
>
> | Dataset |   Base@1  |   ADDER@1 |   Base@3  |   ADDER@3 |   Base@5  |   ADDER@5 |   Base@10 |  ADDER@10|
> | ---------- | ---------- | ---------- | ---------- | ---------- | ---------- | ---------- | ---------- | ---------- |
> | Scifact    |   0.01    |   0.17 |   0.02  |   0.19 |   0.02 |   0.22 |   0.03 |   0.24 |
> | Arguana    |   0.04 |   0.11 |   0.07 |   0.19 |   0.08 |   0.22 |   0.10 |   0.25 |
> | FiQA   |   0.00 |   0.06 |   0.00 |   0.05 |   0.00 |   0.06  |   0.01 |   0.07 |
> | NFCorpus   |   0.03 |   0.07  |   0.02 |   0.07 |   0.02 |   0.07 |   0.03 |   0.06 |
>
> For the same task reported in Table 3, here are the results when using SBert as base: (nDCG@k values)
>
> | Dataset   |   Base@1  |   ADDER@1 |   Base@3  |   ADDER@3 |   Base@5  |   ADDER@5 |   Base@10 |   ADDER@10 |
> | ---------- | ---------- | ---------- | ---------- | ---------- | ---------- | ---------- | ---------- | ---------- |
> |   M   |   0.04    |   0.76    |   0.08    |   0.86    |   0.09    |   0.88    |   0.11    |   0.89 |
> |   Conala  |   0.07    |   0.92    |   0.13    |   0.96    |   0.15    |   0.97    |   0.17    |   0.97 |
> |   Bash    |   0.05    |   0.81    |   0.07    |   0.89    |   0.09    |   0.9 |   0.1 |   0.91 |
> |   SMCalFlow   |   0.01    |   0.95    |   0.02    |   0.98    |   0.02    |   0.98    |   0.03    |   0.98 |

---

> > ### Author Response · Authors · 2023-11-23
> >
> > We hope our response answered the reviewer's questions. As a final step, we are reminding all reviewers to also see the consolidated responses at https://openreview.net/forum?id=n3kFlvVhJM&noteId=UiCjIw5BCv and https://openreview.net/forum?id=n3kFlvVhJM&noteId=HHbXMvYBvT (apart from the individual responses), if they have not already done so.

---

### Official Review · Reviewer_qgfm · 2023-11-06

**Soundness:** 2 fair
**Presentation:** 3 good
**Contribution:** 2 fair
**Rating:** 5
**Confidence:** 4

**Summary:**

This paper proposes an adapter-based method to adapt the existing LLM for information retrieval. A small adapter is trained with a small number of examples. In addition, a transformation is applied to the query representation and to the document representation before calculating their similarity. The proposed method is tested on several small IR datasets as well as code-retrieval datasets. The results show that the adapter-based approach can improve the performance on some of the datasets, while degrades the performance on some others.

**Strengths:**

The paper is relatively well written. The idea of adapter-based approach is well motivated.
The paper presents an interesting idea of adapter for dense IR. Although adapter has been proposed in previous studies for other purposes, it has not been widely used for IR. An adapter can indeed be an interesting solution to create an adapted dense retriever without having to fine-tune a LLM.
The experiments show improvements on some datasets.

**Weaknesses:**

While the idea of adapter is interesting, a key problem of the paper is that it fails to demonstrate that it can improve IR performance. As the experimental results show, the method can only improve on some of the datasets. The advantage of the method is not demonstrated.
The experiments have been carried out on small datasets. The authors argue that retrieval on a small dataset is a particular problem that warrants more explorations. It is unclear why this is the case, and why a retriever that works on large datasets may not be directly applied to small datasets.
The authors also argue that the existing investigations in IR have focused on large ranked lists, and this paper target small ranked lists (considered to be more strict). Again, it is unclear why this size of output is a particular problem. In many previous studies, evaluations have looked at not only tok-K with large K, but also top-K with small K (NDCG@1, NDCG@5, ...). They are not so different from the measures used in this paper. In addition, the paper does not propose a specific method for retrieving only a few documents. So, the difference between large and small K is not so important, and the difference (if any) is not addressed in the paper.
The authors only compared the proposed method with a basic LLM-based retriever. The latter may not be a state of the art of IR. It would be interesting to compare with other baselines such as ANCE and DPR. It would be interesting to also compare the adapter-based method with a fine-tuning-based method. I understand that fine-tuning a LLM with limited examples may not be easy to do or may not be effective. However, to support the argument that adapter is a better solution than fine-tuning, the comparison may be useful.
There may be more analysis of the experimental results. The results are mitigated. Although some possible explanations are provided, one still wonder why improvements/degradations are produced in different cases. Would this be also related to the number of training data?
The concept of heterogeneous datasets is misleading. One could understand that the dataset contains several types of data. The case of NL queries and code dataset is not a real heterogeneous dataset. This is more similar to cross-media retrieval. The proposed method does not seem to be capable of handling truly heterogeneous datasets.

**Questions:**

How does the adapter-based approach compare to a fine-tuning-based method? Would it be possible to provide additional comparison between these methods?
How does it compare to the state-of-the-art IR methods?
What are the specific solutions proposed in the paper for: (1) retrieval on small datasets, (2) for top-K with small K?
What are the general characteristics of NL2X? The description is missing.

---

> ### Author Response · Authors · 2023-11-17
>
> We assume black-box access to the base model. In other words, we do not access to the weights. Hence, adapter-based approach is the only possibility and fine-tuning is not an option.
>
> Regarding comparison to the state-of-the-art IR methods, we note that the general-purpose OpenAI embeddings are a very strong baseline. Looking at the leaderboard on BEIR benchmarks (https://eval.ai/web/challenges/challenge-page/1897/leaderboard/4475), which reports ndcg@10, we see that OpenAI embeddings (our baseline) is very close to the leaders, and even beats the top leader on one dataset (Arguana). For SciFact, we improve over openai embeddings, and thus beat the state-of-the-art (on the ndcg@10 metric). As we discussed in Q1 above, we get higher relative gains on ndcg@1, which is our focus. Further, we explained in the paper why we do not see gains across all benchmarks, and when we don’t, we do not deteriorate the performance of openai by much (and still beat the leaders on some benchmarks, such as Arguana).
>
> The specific solution proposed in the paper is the idea of learning a low-rank residual adapter for embeddings.
>
> We agree with the reviewer that “[our] method can only improve on some of the datasets”. The important point is that we understand when it improves and when it does not. The point is that we should use an adapter only when it is needed.  We are not claiming that one should always attach an adapter for every kind of retrieval task. Only when the base embeddings perform poorly, should we attach an adapter. Our explanation for when adapters are likely to improve, and when not, can guide this choice. The fact that the adapter is not improving the metrics for some benchmarks should be viewed as a learning contribution of our paper. We further note that the improvements due to ADDER are dependent also on the base model, and OpenAI embedding model is a particularly powerful model that is already proficient on retrievals over several different classes of entities. This fact is not true for other base models, such as SBert. ADDERT over SBert consistently improves retrieval performance because SBert embeddings are not “trained” for diverse retrieval tasks; please see the table in the response to Reviewer Tp6G.
>
> Regarding the discussion on large vs small corpus, we only wanted to make the point that retrieval from small corpus is also of interest in certain applications. Of course, one can use retrieval methods that work on large corpus also on small corpus. Similarly for the issue with small versus large K. We wanted to improve metrics @k for k=1,3. As out results show, when Adder improves over baseline, we usually get a higher relative improvement for small k than we get for large k. We want to point out that our Adder approach is almost at par with the state-of-the-art, and even better than the state-of-the-art for SciFact based on the leaderboard for BEIR.
>
> We will be happy to replace the use of “heterogeneous” since it seems it has a connotation different from what we meant by it.

---

> > ### Comment · Reviewer_qgfm · 2023-11-22
> >
> > While OpenAI embedding has performed well on BEIR collections, there are other larger datasets for IR on which OpenAI has not been tested. It would be useful to provide additional baseline results such as ANCE and DPR.
> >
> > The argument of relying on a blackbox system for embedding is reasonable, but one cannot ignore the fact that such an embedding may not be the best choice for IR (unless shown otherwise). There are neural IR models that have been fine-tuned on some IR dataset (namely MSMARCO) and shown to be quite generalized to other IR datasets. Such fine-tuned IR methods could be reasonable alternatives to the general OpenAI embedding.
> >
> > The authors added additional experiments with larger datasets in BEIR. It would still be interesting to test on typical IR datasets such as MSMARCO or TREC datasets.
> >
> > The point that the method is designed for small datasets is still unclear. Why does this method fits more smaller datasets than larger ones, or top-k results with smaller k than larger k?
> >
> > The fact that the method can improve on some datasets and not others may show that the adaptability of the method to each dataset may be poor. The authors argue that one may select to use ADDER or not for each dataset. This would be difficult in practice before running the test of test queries (or validation queries), and there could also be large performance variations among queries.

---

> > ### Author Response · Authors · 2023-11-23
> >
> > We hope the reviewer was able to read our response. As a final step, we are reminding all reviewers to also see the consolidated responses at https://openreview.net/forum?id=n3kFlvVhJM&noteId=UiCjIw5BCv and https://openreview.net/forum?id=n3kFlvVhJM&noteId=HHbXMvYBvT (apart from the individual responses), if they have not already done so.

---

> ### Author Response · Authors · 2023-11-22
>
> We understand the reviewer's concern regarding the OpenAI embeddings not being tested on larger IR datasets. We would like to draw their attention to the following papers:
> - [Text and Code Embeddings by Contrastive Pre-Training](https://arxiv.org/abs/2201.10005): The previous generation of OpenAI embeddings already outperformed ANCE as shown in _Table 5_. _Table 7_ also shows that they generalize well with other large scale IR datasets.
> - [Evaluating Embedding APIs for Information Retrieval](https://arxiv.org/pdf/2305.06300.pdf): This evaluates the predecessors mentioned in the first paper, the latest OpenAI embeddings (`ada-002` which we use in our experiments), and other Embeddings available as a service. The paper finds that the `ada-002` embeddings outperform others in almost all IR tasks on a very wide range of datasets.
> - [Vector Search with OpenAI Embeddings](https://arxiv.org/pdf/2201.10005.pdf): This work builds upon both the previously mentioned works and further extends the evaluation to more datasets like MSMARCO \& TREC-DL19/20.
>
> We hope this addresses the concerns about extensive evaluation of OpenAI Embeddings and their generality over IR benchmarks. An important point to note in all these papers is that the OpenAI embeddings are almost as good as the other leading baselines usually exceeding or trailing them by a very small margin. We would be happy to include results over other baselines in the final version, if the reviewer believes that it would reasonably add more value to this work.
>
>
> We also want to address the concern that the reviewer points out regarding why we perform well on smaller datasets. We discuss this in Section 4.1 of the paper. The proposed technique is more effective in strict retrieval settings given the limited amount of data available and used in the training process of adapters (e.g. Retrieval Augmented Generation). In the limited data setting, we tend to model the training of our adapters with a hard global negatives optimization which helps pick the ideal corpus element for a given query by contrasting it against other corpus elements. This exercise learns better to differentiate between top elements; however, as you go down in the ranking, the relations are not ordered well enough to learn fine differences to prefer one against another. Hence, resulting at better retrievals at smaller `k`.
>
> If the training data was not a constraint, we would be able to finetune/retrain white-box models as well as training dense layers on top of the black-box model which could perform better at all `k` and be able to learn a new embedding distribution for your task. ADDER employs residual adaptation to embeddings; this allows us to massage the inherent distribution of data and not transform it completely. Over larger datasets, we suspect that transformation of the embeddings would be of a much higher degree and denser to learn.
>
> We also obtained experimental results that show the gains from using ADDER and ADDER2 for NL2Code tasks performed using LLMs for where we (a) use ADDER to retrieve few-shot examples and (b) use ADDER to rerank the candidates returned by the LLM. This application was one of our main motivations for designing ADDER and ADDER2. On the Excel's PowerQuery Mashup dataset, we observe that using only OpenAI embeddings gives us a sketch match performance of 0.72, whereas using ADDER gives us a performance of 0.74. As part of ongoing work, we continue to perform experiments on NL2X tasks using ADDER.

---

### Official Review · Reviewer_Tp6G · 2023-11-07

**Soundness:** 2 fair
**Presentation:** 2 fair
**Contribution:** 2 fair
**Rating:** 3
**Confidence:** 3

**Summary:**

The paper proposes to learn a residual term on top of the embedding obtained by a black-box model to steer similarity towards a task-specific notion. The residual term is obtained by a smooth dictionary lookup through a single-layer attention mechanism and trained using pairs of queries and relevant documents. Experiments show that this can improve nDCG over the original embeddings.

**Strengths:**

- The paper discusses the interesting problem of task-specific adaptation of pre-trained general-purpose embeddings with practical relevance due to the increasing popularity of proprietary black-box models.

**Weaknesses:**

- The paper lacks a clear and consistent description of the experimental methodology, and many design choices are motivated at best by anecdotal evidence. In particular:
  - There is no specification of the data sets used to tune the hyperparameters (e.g., the dimensions, but also $\gamma$, which I assume is part of the loss?)
  - It is noted that using batch-local instead of global negative examples leads to worse results, but no performance number is given for this approach.
  - There is a high-level description of a loss function (should be zero if the positive example is closer than the negative), but no concrete instantiation in the form of a formula is given.
  - The subtitle of Table 2 mentions SBert, but its results are not shown further because they are "uniformly worse than those for OpenAI embeddings". It would be interesting (1) to see figures showing this, and (2) to find out if the presented method also works for worse base embeddings.
- The paper lacks a comparison of competing methods, and its distinction from related work is not very precise. In particular, I would expect
  - a baseline that trains the embeddings immediately after initialization with those obtained via the API. My intuition would be that this might lead to worse generalization, but since the evaluation datasets are narrow in domain, this might not be a problem at all.
  - Approaches from the field of learning to rank / neural rerankers, which are state of the art in information retrieval in similarly sized candidate ranking tasks (usually obtained from a first sparse retrieval stage), see e.g. [Craswell22]
  - A comparison with adaptive similarity methods as they are investigated in the field of similarity search, e.g. [Seidl97].

[Craswell22] Nick Craswell et al., "Overview of the TREC 2022 Deep Learning Track", 2022, https://trec.nist.gov/pubs/trec31/index.html
[Seidl97] Seidl, Thomas, and Hans-Peter Kriegel. "Efficient user-adaptable similarity search in large multimedia databases." VLDB. Vol. 97. 1997.

**Questions:**

- The experimental setup seems to contradict itself.
>  We carried out our experiments on regular laptops and desktops, and used no special purpose hardware for training or inference except for the black-box rest API calls to the OpenAI embedding endpoint.

vs.

> To be precise, for our experiments over ADDER & ADDER2, we use a virtual machine with a single Nvidia K80 GPU (with 24GiB of vRAM), 4 CPU cores and 28 GB of RAM

- The tables would be more readable if the numbers of competing approaches were next to each other (e.g. by having column groups for each k=1,3,5,10 value and the approaches next to each other).

- Minor comments:
  - page 7, first paragraph, there is a typo "retrivers"

---

> ### Author Response · Authors · 2023-11-17
> **Response to review by Reviewer Tp6G**
>
> Thanks for pointing out the contradictory description of our experimental setup. Our initial experiments were done on CPU machines, but later when we had to perform multiple runs to generate numbers for our tables, we switched to a virtual machine with GPU. We will fix this error. To provide evidence for the fact that the computational requirement is low, we report here that the time taken to train our ADDER for 1 epoch (containing 886 batches with batch size 16) is 8.97s on GPU, whereas the same took 71.73s on CPU. So, the experiments are doable on CPU, they just take a little longer but still reasonable time.
>
> Thanks also for the suggestion to rearrange the columns of the tables. We will do so.
>
> Regarding details of the datasets used and hyperparameters, we make use of the train datasets given as a part of BEIR. In case the associated training set was not available, we sampled 20% of the data from the test set for training and used the remaining 80% for test. This also highlights once again that our method does not need a lot of data to learn a general solution. Our hyperparameters were chosen by performing a grid search. Further, a smaller initial learning rate was found to be better for larger datasets. The $\gamma$ found most optimal was 0.1 using a step size of 30 epochs.
>
> Regarding batch local negatives vs global negatives, we observed that using local negatives reduces performance over the base embeddings by over 10% or more over the datasets. Since we saw this in initial experimentation, we did not record the exact numbers. We would be happy to perform those experiments and report the numbers.
>
>  The loss function is as follows: if q, c, and c’ are the transformed embeddings of the query, true positive corpus element, and the global negative corpus element, then the loss l(q,c,c’) = max(0, dot(q,c’) - dot(q,c)), where dot is the cosine similarity function.
>
>  We thank the reviewer for their comment on SBert, and include the following results for SBERT for both the default and adapted embeddings scenarios:
>
>  For the same BEIR benchmarks reported in Table 2, here are the results with using SBert as base. (nDCG@k values)
>
> | Dataset |   Base@1  |   ADDER@1 |   Base@3  |   ADDER@3 |   Base@5  |   ADDER@5 |   Base@10 |  ADDER@10|
> | ---------- | ---------- | ---------- | ---------- | ---------- | ---------- | ---------- | ---------- | ---------- |
> | Scifact | 0.01 |   0.17 |   0.02  | 0.19 | 0.02 | 0.22 | 0.03 | 0.24 |
> | Arguana |   0.04 |   0.11 |   0.07 | 0.19 | 0.08 | 0.22 | 0.10 | 0.25 |
> | FiQA | 0.00 |   0.06 |   0.00 |   0.05 | 0.00 | 0.06 | 0.01 | 0.07 |
> | NFCorpus | 0.03 | 0.07 | 0.02 | 0.07 | 0.02 | 0.07 | 0.03 | 0.06 |
>
>  For the same task reported in Table 3, here are the results when using SBert as base: (nDCG@k values)
>
> | Dataset   |   Base@1  |   ADDER@1 |   Base@3  |   ADDER@3 |   Base@5  |   ADDER@5 |   Base@10 |   ADDER@10 |
> | ---------- | ---------- | ---------- | ---------- | ---------- | ---------- | ---------- | ---------- | ---------- |
> |   M   |   0.04    | 0.76  |   0.08    |   0.86    |   0.09    |   0.88    |   0.11    |   0.89 |
> |   Conala  | 0.07  | 0.92  |   0.13    |   0.96    |   0.15    |   0.97    |   0.17    |   0.97 |
> |   Bash    | 0.05 |   0.81    |   0.07    |   0.89    |   0.09    |   0.9 |   0.1 |   0.91 |
> |   SMCalFlow   |  0.01 | 0.95 | 0.02 | 0.98 | 0.02 | 0.98 | 0.03 | 0.98 |
>
> Regarding comparison to competing methods, we observed that OpenAI embeddings were giving state-of-the-art results in most cases; in some cases, OpenAI embeddings performed even better than the leader on the BEIR leaderboard. That was the reason to start with OpenAI embeddings. The ADDER adapter works over those embeddings, and even improves over them in some cases.
>
> We also realize that although we did a lot of experiments with other modeling choices, we only reported the numbers with the best architecture we found. In a revised version, we will include performance numbers for following ablations: (1) not using residual adaptation and instead using direct MLP adapter, (2) not using global hard negatives and instead using batch negatives, (3) not using contrastive loss.
>
> We also thank the reviewer for the interesting [Seidl97] reference that we will discuss in the revised version.

---

> > ### Comment · Reviewer_Tp6G · 2023-11-22
> >
> > Thank you for your response. Some parts have been resolved, but I comment below on the parts that remain open. I encourage the suggested revisions and am open to adjusting my rating if I see an improvement.
> >
> > >Regarding details of the datasets used and hyperparameters, we make use of the train datasets given as a part of BEIR. In case the associated training set was not available, we sampled 20% of the data from the test set for training and used the remaining 80% for test. This also highlights once again that our method does not need a lot of data to learn a general solution. Our hyperparameters were chosen by performing a grid search. Further, a smaller initial learning rate was found to be better for larger datasets. The found most optimal was 0.1 using a step size of 30 epochs.
> >
> > I am still not sure whether I correctly understand your hyperparameter selection:
> > > Our hyperparameters were chosen by performing a grid search
> >
> > Which metric did you optimize here? nDCG@1? And on what part did you evaluate? On the 80% test part? If so, how would this be different from directly optimizing the hyperparameters on the final test set?
> >
> > > We thank the reviewer for their comment on SBert, and include the following results for SBERT for both the default and adapted embeddings scenarios:
> >
> > Thanks for providing these results. They seem more convincing to me, especially for the task in Table 3, where they show a large improvement over the baseline.

---

> > > ### Author Response · Authors · 2023-11-23
> > >
> > > Thanks for approving the suggested revisions, and we will now go ahead and prepare the revision. Unfortunately, due to time constraints, we are unable to upload the revised version before the rebuttal period ends. Our ablation experiments are still running.
> > >
> > > The reviewer is correct: we optimize on the ndcg@1 performance. Our loss function optimizes on the having the correct corpus embedding closest to the adapted embedding.
> > >
> > > No, we do not perform evaluation (for hyperparameter selection) on the full, or even 80%, of the test set. It is always on the dev set. In case of one benchmark set (Arguana), the train+dev set was picked to be 20% of the test set (because Arguana had no train set).
> > >
> > > For the NL2X tasks we had access to well defined train and test sets.
> > >
> > > We also remind all reviewers to also see the consolidated responses at https://openreview.net/forum?id=n3kFlvVhJM&noteId=UiCjIw5BCv and https://openreview.net/forum?id=n3kFlvVhJM&noteId=HHbXMvYBvT (apart from the individual responses), if they have not already done so.

---

### Official Review · Reviewer_j6bh · 2023-11-09

**Soundness:** 3 good
**Presentation:** 3 good
**Contribution:** 3 good
**Rating:** 6
**Confidence:** 4

**Summary:**

This work addresses the problem of highly accurate retrieval of a small number of text documents from a relatively small corpus when the query and the corpus are from different domains. It is motivated by the recent interest in retrieval-augmented generation applications involving LLMs where a small number of highly relevant documents are used to provide additional context to the generative task specified by the query. The work argues that classical document retrieval methods are not useful for such applications due to domain mismatch between the query and the corpus, task specificity of retrieval and stricter requirement for retrieval efficacy in the top retrieval results. The work assumes that in such applications, the target corpus is of small size (a few thousand documents) and that a limited number of examples of "good" retrievals (a few thousand) are available for this corpus.

It proposes a solution to the problem that consists of transforming pretrained embeddings for the query and corpus documents. Specifically, the work proposes to do a parameter efficient finetuning of pretrained embeddings without accessing the weights of the pretrained embedding model and using only a small sized tuning data set that consists of examples of "good" retrievals. Towards this the work proposes augmenting the pretrained embedding model with small adapter models that can learnt from the small tuning data set. Given the query (document), the adapters compute a perturbation vector by looking up a learnable dictionary which is then added to the pretrained embedding of the query (document).

The dictionary used for softmaxed lookup is small in size and learnt from the tuning data set (consisting of examples of "good" retrievals). The dictionary is learnt by enforcing that the perturbed/adapted embedding of q is closer to that of c than to global hard negatives for q in the corpus.

The work presents results from an experimental study to evaluate the value added by the proposed method on  five relatively small BEIR Datasets. It employs OPENAI ADA embeddings as the baseline retrieval system as well as the pretrained black-box embedding for adaptation. Retrieval efficacy is measured using NDCG @1, 3, 5 and 10. Though adaptation seems to improve retrieval performance in some cases, there is no clear winner. In some cases adapting only query embedding seems to be better, in some other cases adapting both query and corpus embeddings seems to be better (SciFact) and in other cases not adapting seems to be better (FiQA). The differences in gains are attributed to the differences in alignment of the notion of semantic similarity learnt by OPENAI ADA with what is intended in the benchmarks." However, there is no detailed analysis of the errors to validate this hypothesis.

The work also presents results from another experimental study on retrieval tasks involving natural language queries and corpus of code fragments. The gains are impressive on one dataset (SMCALFLOW), positive but marginal on another (BASH) and negative on the third (CONALA).

**Strengths:**

1. The problem of task specific fine tuning of retrieval models is very interesting and important. The specific setting in which the problem is being attempted to solve, i.e. treating pretrained embedding model as black box is also interesting. The line of attack using learnable dictionary-based light weight adapters is very interesting. It doesn't need access to the weights of the pretrained embedding model and pretraining embedding model need not be retrained.

2. Adaptation is not computationally intensive and can be done on simple hardware.

3. Adaptation can be attempted with relatively small sized tuning data set.

**Weaknesses:**

1. Though the pretrained embedding model is treated as a black box, adaptation is still strictly tied to the specific pretrained embedding model and needs to be done separately for each pretrained embedding model. As the pretrained embedding model changes over time due to retraining/continual updating, the adapters also need to be retrained.

2. The work assumes that in many applications, the target corpus is of small size (a few thousand documents) and that a limited number of examples of "good" retrievals (a few thousand) are available for this corpus. This is however a restrictive assumption. Typically the corpus is several orders larger in size than the available number of "good" retrievals.

3. Improvements in retrieval efficacy is not impressive. The proposed approach is not a clear winner always. It is also clear what is the best improvement one can hope to get from this line of attack. Is the lack of large training data sets the major hurdle for significant further improvement or there are inherent limitations in the methodology? Even if large amount of training data were available, the global hard negatives idea employed would not work as is.

4. There is no study of the improvement brought by the proposed adapter-based retrieval approach to RAG tasks.

**Questions:**

1. What is the architecture for the adders?


2. "current state of the adapted model" -> art


3. Several references are repeated.


4. "The typical values we use for h range from 16 to 128."

5. What is the value of h in the experiments? Is it fixed for all data sets? How was the value chosen?

---

> ### Author Response · Authors · 2023-11-17
> **Response to review by Reviewer j6bh**
>
> The reviewer is correct when they state that the adaptation is tied to the specific blackbox model. When the blackbox embedding model changes, the adapters also need to be retrained. We would like to point out that even though this is true, there is minimal effort in retraining the adaptors as compared to other techniques like finetuning. For 1 epoch consisting of 668 batches (batch size 16) our technique only takes ~71 seconds on the largest dataset FiQA-2018 (corpus size 57k), that too on a CPU only compute (on a single K-80 GPU this takes only 8 secs). Hence the effort in training adapters is very minimal and not compute-intensive.
>
> We agree with the reviewer that working on only small corpus sizes is restrictive. We do report numbers for FiQA and NFCorpus, where the corpus sizes are larger, and the number of good retrievals is less. There are applications where corpus sizes are small and training data is sparse, which serve as motivation for this work.
>
> We agree that we cannot bring about large improvements since our technique employs a residual approach. It is limited to adapting the embeddings and not changing them entirely. Further, we envision our method to be employed in cases specifically where there is a scarcity of training data and larger models might be overfit to this. In case there is sufficient training data one can finetune a whitebox model or learn a denser layer on top of the blackbox embedding and would see better improvements since the inherent quality of dense embeddings would improve. Another benefit of our approach is the low computational requirement, which allows one to do online learning to tune ADDERs to handle new data/ outliers. Finally, the improvements in retrieval efficacy are dependent on the base model -- we observed significantly more pronounced gains SBert; please see the table in comment to Reviewer LpK4.
>
> Thank you for asking about results on RAG. We report results on the M dataset proposed in [1], we would be happy to include these results in the final version of the paper as well. We observe the using only OpenAI embeddings gives us a sketch match performance of 0.72 on the M dataset. Whereas when we use ADDER to select the fewshot prompt and perform reranking over generated candidates, we obtain a Top-1 performance of 0.74.
>
> [1] Anirudh Khatry, Joyce Cahoon, Jordan Henkel, Shaleen Deep, Venkatesh Emani, Avrilia Floratou, Sumit Gulwani, Vu Le, Mohammad Raza, Sherry Shi, Mukul Singh, Ashish Tiwari. From Words to Code: Harnessing Data for Program Synthesis from Natural Language (arXiv:2305.01598)

---

> > ### Comment · Reviewer_j6bh · 2023-11-20
> > **Rebuttal**
> >
> > I thank the authors for their rebuttal and specifically for answering my question on RAG experiments. As the result is encouraging, I suggest the authors to consider doing a more extensive study of the applications of the proposed approach in RAG setting to strengthen the current work.

---

> > > ### Author Response · Authors · 2023-11-23
> > > **Reminder to also see the consolidated responses.**
> > >
> > > As a final step, we are reminding all reviewers to also see the consolidated responses at https://openreview.net/forum?id=n3kFlvVhJM&noteId=UiCjIw5BCv and https://openreview.net/forum?id=n3kFlvVhJM&noteId=HHbXMvYBvT (apart from the individual responses), if they have not already done so.

---

### Official Review · Reviewer_LpK4 · 2023-11-10

**Soundness:** 3 good
**Presentation:** 3 good
**Contribution:** 2 fair
**Rating:** 5
**Confidence:** 3

**Summary:**

The paper focuses on heterogeneous strict retrieval that may have increased importance in the context of Retrieval Augmented Generation (RAG) with Large Language Models (LLMs). The retrieval is heterogeneous in the sense that the query may be in natural language while the corpus may contain code, documentation, structured, and semi-structured text, or artifacts. The retrieval problem is also said to be strict because of the increased importance of a few top candidates. The strict retrieval may be important because of the limited context window and positional bias of the LLMs in RAG. The proposed method involves using adapters that perturb the general-purpose embeddings of the query (and in one variation, the corpus) to achieve task-specific improvements. The adapters, whose parameters can be learned using a small task-specific training dataset, add residual terms to the lower rank of the general-purpose embeddings for task-specific adaptation. The parameters of the adapters are learned with a contrastive loss. The proposed approach is compared with general-purpose LLM-based embeddings on classical and heterogeneous retrieval tasks.

**Strengths:**

1. The paper proposes task-specific adaptation of general-purpose embeddings using low-rank adaptation, which can improve training efficiency and works with little task-specific data. The proposed approach shows some improvements in certain tasks on both classical and heterogeneous retrieval.
2. The authors formulate a new problem named heterogeneous strict retrieval and explain why it may be an important aspect of LLM-based RAG.
3. The proposed approach has been clearly described with extensive details, which potentially makes the work reproducible for the community.

**Weaknesses:**

1. One limitation of the paper is the lack of baselines with task-specific adaptation. The only baseline shown in the evaluation is based on general-purpose embeddings, which may not be a fair comparison.
2. The paper claims to enable improved heterogeneous and strict retrieval, but it was not clear to me from the experimental results whether the improvement in strict retrieval has been substantiated with evidence.
3. The improvements resulting from the proposed methodology appear limited in some instances, and it even underperforms compared to the baseline in the cases of FiQA and CONALA.

**Questions:**

1. How does the proposed approach improve strict retrieval compared to baselines? How do the experimental results support this improvement?
2. How is the dimension size of h selected? It was mentioned that h ranges from 16 to 128, but I could not find a discussion on this in the experiments. What are the pros and cons of choosing higher or lower values for h?

---

> ### Author Response · Authors · 2023-11-17
> **Response to Review by Reviewer LpK4**
>
> Q1. We claimed that our approach, Adder, improves strict retrieval based on the evidence that the relative improvement on ndcg@1 metric is mostly greater than the relative improvements we get on ndcg@10 (and ndcg@k for larger k). This can be verified for the data presented in Table 2 and 3. For example, in Table 3, Row 1 for smcalflow, we see a 50% relative improvement on ndcg@1 and a smaller 30% relative improvement on ndcg@10. In general, we observed that whenever there are gains, the relative gains on ndcg@k are higher for a smaller k. That is why we said that our approach is more suited for strict retrieval tasks. We will suitably justify our claims in the revised version.
>
> Q2: We treat h, which is the rank of the LoRA transformation matrix, as a hyperparameter in our experiments. We selected the value of h from the set [16, 32, 64, 128, 256, 512, 768, 1024] through validation over the train/dev sets. We observed that the best values for h across the datasets we reported emerged to be around 256 (for ADDER) & 512 (for ADDER2). We also observed some robustness to the choice of h in our experiments – the reported metrics did not change very significantly when varying h in a range centred around the ‘best’ h determined by validation over the available data.
>
> Weakness1: General-purpose OpenAI embeddings are very strong baselines. Looking at the leaderboard on BEIR benchmarks (https://eval.ai/web/challenges/challenge-page/1897/leaderboard/4475), which reports ndcg@10, we see that OpenAI embeddings (our baseline) is very close to the leaders, and even beats the top leader on one dataset (Arguana). For SciFact, we improve over openai embeddings, and thus beat the state-of-the-art (on the ndcg@10 metric). As we discussed in Q1 above, we get higher relative gains on ndcg@1, which is our focus. Further, we explained in the paper (and Weakness 3, below) why we do not see gains across all benchmarks, and when we don’t, we do not deteriorate the performance of openai by much (and still beat the leaders on some benchmarks, such as Arguana).
>
> Weakness 2 about strict retrieval is the same point as Q1.
>
> Weakness 3: It is true that our approach does not uniformly improve performance across all benchmarks. However, there is good reason for it, as we discuss in the last paragraph of Section 4.1 and the last two paragraphs of Section 4.2. We believe that knowing when our approach improves over the baseline, and when it does not, is a strength of our work. It informs us on when to use an embedding adapter or when not. So, while the first impression from our results (Tables 2,3) may be that our approach does not work consistently, if we look deeper at the class of benchmarks where it does not improve over the base openai embeddings, one realizes that the results validate our hypothesis – use adapters when the retrieval task has nuances (code, semi-structured text) that the base model is not likely to exploit.
>
> Given that the OpenAI embeddings are widely trained on public data, including language and code, it is possible that they were also trained on some of the tasks/data from the BEIR dataset. That can make it difficult to further improve upon the OpenAI embeddings by tuning them on the same data. On the other hand, a weaker embedding model, like sBERT,  provides more opportunity for improvement by tuning it for the task. In the results presented below for SBert + Adder, we do see a consistent gain across all IR tasks ranging from 2x to 10x in nDCG@1 across datasets. Thus, the overall gains that we can expect from our approach depends on the initial pretraining of the underlying model.
>
> For the same BEIR benchmarks reported in Table 2, here are the results with using SBert as base. (nDCG@k values)
>
> | Dataset |   Base@1  |   ADDER@1 |   Base@3  |   ADDER@3 |   Base@5  |   ADDER@5 |   Base@10 |  ADDER@10|
> | ---------- | ---------- | ---------- | ---------- | ---------- | ---------- | ---------- | ---------- | ---------- |
> | Scifact | 0.01 |   0.17 |   0.02  | 0.19 | 0.02 | 0.22 | 0.03 | 0.24 |
> | Arguana |   0.04 |   0.11 |   0.07 | 0.19 | 0.08 | 0.22 | 0.10 | 0.25 |
> | FiQA | 0.00 |   0.06 |   0.00 |   0.05 | 0.00 | 0.06 | 0.01 | 0.07 |
> | NFCorpus | 0.03 | 0.07 | 0.02 | 0.07 | 0.02 | 0.07 | 0.03 | 0.06 |
>
> For the same task reported in Table 3, here are the results when using SBert as base: (nDCG@k values)
>
> | Dataset   |   Base@1  |   ADDER@1 |   Base@3  |   ADDER@3 |   Base@5  |   ADDER@5 |   Base@10 |   ADDER@10 |
> | ---------- | ---------- | ---------- | ---------- | ---------- | ---------- | ---------- | ---------- | ---------- |
> |   M   |   0.04    | 0.76  |   0.08    |   0.86    |   0.09    |   0.88    |   0.11    |   0.89 |
> |   Conala  | 0.07  | 0.92  |   0.13    |   0.96    |   0.15    |   0.97    |   0.17    |   0.97 |
> |   Bash    | 0.05 |   0.81    |   0.07    |   0.89    |   0.09    |   0.9 |   0.1 |   0.91 |
> |   SMCalFlow   |  0.01 | 0.95 | 0.02 | 0.98 | 0.02 | 0.98 | 0.03 | 0.98 |

---

> > ### Comment · Reviewer_LpK4 · 2023-11-21
> > **Rebuttal Response**
> >
> > Thanks to the authors for clarifying the queries regarding strict retrieval and hyper-parameter selection.

---

> > > ### Author Response · Authors · 2023-11-23
> > >
> > > Thank you for acknowledging the response. As a final step, we are reminding all reviewers to also see the consolidated responses at https://openreview.net/forum?id=n3kFlvVhJM&noteId=UiCjIw5BCv and https://openreview.net/forum?id=n3kFlvVhJM&noteId=HHbXMvYBvT (apart from the individual responses), if they have not already done so.

---

### Author Response · Authors · 2023-11-17
**Consolidated response to the reviews**

Multiple reviewers noted from the experimental tables that ADDER and ADDER2 do not consistently improve retrieval performance. We wanted to make two important points about our results.
- First, the general-purpose OpenAI embeddings are a very strong baseline. Looking at the leaderboard on BEIR benchmarks (https://eval.ai/web/challenges/challenge-page/1897/leaderboard/4475), which reports nDCG@10, we see that OpenAI embeddings (our baseline) is very close to the leaders, and even beats the top leader on one dataset (Arguana). For SciFact, we improve over the OpenAI embeddings, and thus beat the state-of-the-art (on the reported nDCG@10 metric).
- Second, ADDER and ADDER2 adaptations should be used only when the retrieval task has nuances (code, semi-structured text) that the base model is not likely to be able to exploit. OpenAI embeddings can exploit several features in semi-structured text and code. That is why we don't see consistent improvements across all benchmarks (as we have also explained in detail in Section 4.1 and 4.2 in the paper.) A weaker embedding model, such as SBert, is unable to do the same, and hence, we expect ADDER and ADDER2 to more consistently improve performance when SBert is the base model. In fact, our experiments confirmed that hypothesis, and we see orders of magnitude relative gains over SBert when using ADDER and ADDER2, as shown below:

ADDER over SBert embeddings on selected BEIR datasets (cf. Table 2 in the paper that did the same for OpenAI embeddings)
| Dataset | Base@1 | ADDER@1 | ADDER2@1 | Base@3 | ADDER@3| ADDER2@3 | Base@5 | ADDER@5 | ADDER2@5 | Base@10 | ADDER@10| ADDER2@10|
| --- | --- | --- | --- | --- | --- | --- | --- | --- | ---| ---| ---| --- |
| Scifact | 0.01 | 0.17 | 0.35 | 0.02 | 0.19 | 0.42 | 0.02 | 0.22 | 0.44 | 0.03 | 0.24 | 0.46 |
| Arguana | 0.04 | 0.11 | 0.13 |0.07 | 0.19 | 0.21 | 0.08 | 0.22 | 0.24 | 0.10 | 0.25 | 0.27 |
| FiQA | 0.00 | 0.06 | running | 0.00 | 0.05 | running | 0.00 | 0.06 | running | 0.01 | 0.07 | running |
| NFCorpus | 0.03 | 0.07 | 0.05 | 0.02 | 0.07 | 0.05 | 0.02 | 0.07 | 0.05 | 0.03 | 0.06 | 0.05|

(ADDERx@k means nDCG@k for ADDERx; running indicates the results will be available shortly)

ADDER over SBert embeddings on NL2X tasks (cf. Table 3 in the paper that did the same for OpenAI embeddings)
| Dataset | Base@1 | ADDER@1 | Base@3 | ADDER@3 | Base@5 | ADDER@5 | Base@10 | ADDER@10 |
| ---------- | ---------- | ---------- | ---------- | ---------- | ---------- | ---------- | ---------- | ---------- |
| M | 0.04 | 0.76 | 0.08 | 0.86 | 0.09 | 0.88 | 0.11 | 0.89 |
| Conala | 0.07 | 0.92 | 0.13 | 0.96 | 0.15 | 0.97 | 0.17 | 0.97 |
| Bash | 0.05 | 0.81 | 0.07 | 0.89 | 0.09 | 0.9 | 0.1 | 0.91 |
| SMCalFlow | 0.01 | 0.95 | 0.02 | 0.98 | 0.02 | 0.98 | 0.03 | 0.98|

(ADDER2 results for NL2X are still running)

Multiple reviewers also asked for more ablation experiments. We will include performance numbers for following ablations in the final version: (1) not using residual adaptation and instead using direct MLP adapter, (2) not using global hard negatives and instead using batch negatives, (3) not using contrastive loss. In the process of developing ADDER and ADDER2, we had performed these experiments to guide our design choices. We saw performance gains when we added each component in our final architecture.

We also obtained experimental results that show the gains from using ADDER and ADDER2 for NL2Code tasks performed using LLMs where we (a) use ADDER to retrieve few-shot examples and (b) use ADDER to rerank the candidates returned by the LLM. This application was one of our main motivations for designing ADDER and ADDER2. On the Excel's PowerQuery Mashup dataset, we observe that using only OpenAI embeddings gives us a sketch match performance of 0.72, whereas using ADDER gives us a performance of 0.74. As part of ongoing work, we continue to perform experiments on NL2X tasks using ADDER.

---

> ### Author Response · Authors · 2023-11-22
> **Supporting Qualitative artifacts for End-To-End evaluation for ADDER on NL to Power Query M**
>
> To illustrate why we need to adapt or tune a general-purpose embedding, consider the Natural Language (NL) utterance _Select all unique rows_ and the utterance _Remove duplicates from the table_. As NL sentences, these two are different: one is talking about picking elements, while the other is talking about removing elements. However, both these NL utterances map to the same code in any NL2Code task. When a model trained on sentence similarity is used to compute similarity, we are likely to get a very low similarity score for these two sentences; in fact, OAI embeddings give a 0.1 score here indicating that the two sentences are _not_ similar. In contrast, Adapted Embeddings (ADDER) show that these two sentences are very similar by producing a score of 0.9.  Thus, ADDER is able to adapt the embeddings so that it corresponds to the _code similarity_ intent when we are dealing with NL2Code tasks.
>
> **Motivating example from Few-Shot Selection**
>
> Let us say we are given the task of generating the code given below from the following NL. \
> \
> NL: _only keep the 10 longest questions_
>
> Code: ``Table.FirstN(Table.Sort(Table1,{"Length"},Descending),10)``
>
> The user wants to retain the 10 longest questions in the table.
> Default openai embeddings fixate on the word _question_ and the number _10_. As a result if we use OpenAI embeddings to retrieve few-shot examples, we see retrieved samples such as \
> \
> NL: _Combine the two questionnaires into one table._
>
> Code: ``Table.Combine({#"Questionaire 1", #"Questionaire 2"})`` \
> \
> NL: _Retrieve the first 10 rows from Table1._
>
> Code: ``Table.FirstN(Table1, 10)`` \
> \
> NL: _Select all rows from Table1 where the difference between the EndDate and StartDate is 10 days._
>
> Code: ``Table.SelectRows(Table1,
>             each Duration.Days([EndDate]-[StartDate]) = 10
>             )``
>
> \
> ADDER rather gives the following example:\
> \
> NL: _Sort the table1 by columns C%, d%, and r% and return the first 10 rows._
>
> Code: ``Table.FirstN(
>                 Table.Sort(Table1,{"C%","d%","r%"}),10
>                 )``\
> \
> The sample retrieved using ADDER captures the _code similarity_ intent much better because the code here first sorts the column in a given order and then picks the top 10 from the sorted list, which is what the code for user's NL query should also do.\
> \
> As a second example, consider the task where the user provides the following NL utterance, and expects the model to generate the subsequent code.\
> \
> NL: _if column "URL" does not end with ".html" then I want to append the text ".html" to the column_
>
> Code: ``Table.TransformColumns(Table1,
>             {{"URL",
>             each
>                 if Text.EndsWith(_, ".html")
>                 then _
>                 else _ & ".html",
>             type text}})``\
> \
> Open AI Embeddings selects samples that contains string constants within quotes, as in the NL utterance:\
> \
> NL: _Add a new column to Table1 called "End String" and fill it with the text rel="nofollow">" followed by the value of the API column and ending with "</a></td>".
>
> Code: ``Table.AddColumn(Table1, "End String",
>                 each
>                     """ rel=""nofollow"">" & [API] & "</a></td>",
>                 type text) ``\
> \
> NL: Replace the value of the URL column in Table1 with the text "https://ir.tevapharm.com/news-" whenever the value contains "/news-".
>
> Code: ``Table.ReplaceValue(Table1,"/news-",
>                         "https://ir.tevapharm.com/news-",
>                         Replacer.ReplaceText,{"URL"})``\
> \
> ADDER gives the following examples:\
> \
> NL: _Add a column to Table1 called Right, and for each row, if the Header ends with the letter W, then the value in the Right column should be W, otherwise it should be null and the type should be text.\
> \
> Code: ``Table.AddColumn(Table1, "Right",
>                         each
>                             if Text.EndsWith([Header],"W")
>                             then "W" else null,
>                         type text)``\
> \
> NL: _Transform the columns of Table1 by replacing the href column with the text before the delimiter ">".
>
> Code: ``Table.TransformColumns(Table1,
>                             {{"href", each
>                                 Text.BeforeDelimiter(_, ">"),
>                             type text}})``\
> \
> Note how __EndsWith__ and __Table.TransformColumn__ both have been covered in the samples selected by ADDER.

---

### Meta-Review · Area_Chair_UiCB · 2023-12-06

**Metareview:**

Retrieving from heterogenous data is an important problem, we have seen multimodal RAG gaining more and more traction.
This paper presents a light-weight method that works with embeddings of black-box LLMs -  it uses adapters to modify general-purpose embeddings for task-specific improvements  - an interesting idea.
Despite some positives, the paper will be strengthened by comprehensively addressing the following issues:  lack of fair baselines, variable performance across datasets, this inconsistency raises questions about the method’s reliability and general applicability.

**Justification For Why Not Higher Score:**

Despite some positives, the paper will be strengthened by comprehensively addressing the following issues:  lack of fair baselines, variable performance across datasets, this inconsistency raises questions about the method’s reliability and general applicability.

**Justification For Why Not Lower Score:**

N/A

---

### Decision · Program_Chairs · 2024-01-16

Reject